# Two-Thirds Dose Photodynamic Therapy for Pachychoroid Neovasculopathy

**DOI:** 10.3390/jcm10102168

**Published:** 2021-05-17

**Authors:** Koji Tanaka, Ryusaburo Mori, Yu Wakatsuki, Hajime Onoe, Akiyuki Kawamura, Hiroyuki Nakashizuka

**Affiliations:** Department of Ophthalmology, Nihon University School of Medicine, Tokyo 101-8309, Japan; mori.ryusaburo@nihon-u.ac.jp (R.M.); wakatsuki.yu@nihon-u.ac.jp (Y.W.); onoe.hajime@nihon-u.ac.jp (H.O.); kw-eye-c@xb3.so-net.ne.jp (A.K.); nakashizuka.hiroyuki@nihon-u.ac.jp (H.N.)

**Keywords:** PNV, OCT angiography, 2/3PDT

## Abstract

Pachychoroid neovasculopathy (PNV) is treated with antivascular endothelial growth factor (VEGF) injection and photodynamic therapy (PDT), but no curative treatment has yet been established. We aimed to clarify the treatment results of a reduced dose of PDT for PNV. The subjects were 27 eyes of 27 patients (male:female = 20:7, mean age 58.9 years). PDT, at 2/3 of the conventional dose (2/3PDT), was administered once. The patients were then observed for one year. Eyes with polypoidal choroidal vasculopathy (PCV) were excluded. We investigated the associations among the central retinal thickness, choroidal thickness, and visual acuity changes before treatment and one, three, six and 12 months after PDT. When serous retinal detachment was increased or unchanged or new hemorrhages were observed, as compared with pretreatment findings, intravitreal injection of an anti-VEGF agent was performed. Visual acuity was significantly improved, as compared to before treatment, at three, six, and 12 months after 2/3PDT. Foveal retinal thickness was significantly decreased after versus before treatment in the 2/3PDT group (*p* < 0.001). Foveal choroidal thickness was also significantly reduced in the 2/3PDT group (*p* = 0.001). Additional intravitreal anti-VEGF agent injections were administered to three patients (11%), while 24 (89%) required no additional treatment during the one-year follow-up period. For PNV without polyps, 2/3PDT appears to be effective.

## 1. Introduction

Pachychoroid neovasculopathy (PNV) was proposed by Freund et al. and is characterized by a thickened choroid, dilated choroidal large vessels, few drusen, choroidal vascular hyperpermeability (CVH) in the late phase of indocyanine green fluorescein angiography (ICGA), and choroidal neovascularization (CNV) [1,2]. PNV also includes polypoidal choroidal vasculopathy (PCV) [3], but aflibercept and photodynamic therapy (PDT) have been shown to be effective for treating PCV [4,5]. Treatment of PCV-containing PNV was reported [6]. The efficacy of a combination of half-dose PDT and aflibercept for PNV without polyps was also recently described [7]. Another recent study focused on full-dose PDT combined with ranibizumab for PNV without polyps [8]. In Europe and the USA, Smretschnig et al. reported on the treatment of CNV after chronic central serous chorioretinopathy (CSC), a clinical entity similar to PNV [9]. Furthermore, in Asia, the rate of dry macula was higher in eyes treated with aflibercept than in those treated with ranibizumab for PNV [10]. PNV also reportedly tends to have longer intervals before intravitreal injections than typical age-related macular degeneration (tAMD) [11]. On the other hand, anterior chamber antivascular endothelial growth factor (VEGF) levels in PNV were noted to be lower than those in tAMD [12], thereby rendering anti-inflammatory drugs less effective. Lee et al. examined the one-year results for full-dose PDT for Type 1 CNV with a thickened choroid (similar to PNV) and found that 40% of eyes showed relapse at one year [13]. Despite the various studies conducted to date, no optimally effective treatments for PNV have yet been established. Given these background factors, we examined the potential usefulness of a two-thirds dose of PDT (2/3PDT) for PNV. We retrospectively investigated the clinical outcomes of eyes with PNV, excluding PCV, receiving this 2/3PDT treatment.

## 2. Materials and Methods

### 2.1. Subjects

Twenty-seven patients (20 men, seven women; mean age 58.9 years) were enrolled at Nihon University Hospital in Tokyo between Jun 2017 and March 2019. PNV was defined as subfoveal CNV on optical coherence tomography angiography (OCTA), macular pachyvessels and serous retinal detachment (SRD) on OCT, and increased choroidal vascular hyperpermeability (CVH) on ICGA. All the diagnoses were determined by two ophthalmologists (K.T., R.M.). The treatment outcomes of 27 eyes of these patients who underwent 2/3PDT were followed up for one year. The patients received a verteporfin injection at 4 mg/m^2^ body surface area and then underwent PDT with a light fluence of 50 J/cm^2^ using a Visulas PDT system 690S (Carl Zeiss Japan) for 83 s. The greatest linear dimension (GLD) of the PNV lesion was determined employing ICGA. Cases with PCV or subretinal hemorrhage, suspected secondary CNV due to inflammatory disease, past vitreous surgery, or high myopia (>−6 diopter) were excluded.

The central macular thickness (CMT) and central choroidal thickness (CCT) were measured by EDI-OCT (SPECTRALIS HRA–OCT, Heidelberg Engineering Inc.).

OCTA was performed using the RTVue XR Avanti with Angio Vue (Optovue Inc. Fremont, CA, USA). Macular cubes (3 × 3 mm) were acquired to detect the presence of CNV. Therapeutic effects were defined as no SRD on OCT, absence of hemorrhage as “dry macula”. We investigated the associations between central retinal thickness, choroidal thickness, and visual acuity changes before treatment with those one, three, six and 12 months after 2/3PDT. Additional treatment consisted of an intravitreal injection of aflibercept if SRD was increased or unchanged or if new hemorrhages were observed that had not been detected prior to treatment. Macular atrophy was defined as the presence of a black site indicative of a defect in the retinal pigment epithelium with fundus autofluorescence after treatment.

All patients provided informed consent and we performed this study, which complied with the guidelines of the Declaration of Helsinki, with approval from our institutional review board.

### 2.2. Statistical Analysis

Data are presented as means ± SD. The paired t-test was performed to assess changes in visual acuity, foveal retinal thickness, and choroidal thickness after versus before treatment. Logistic regressions were performed for factors in patients who required additional treatment. A value of *p* < 0.05 was considered to indicate a statistically significant difference. SPSS ver.26 (IBM Corp.) was used for these analyses.

## 3. Results

The clinical features of participants are shown in Table 1. Early Treatment Diabetic Retinopathy Study (ETDRS) visual acuity was not significantly increased one month after treatment but was significantly increased at the measurements from three to 12 months after 2/3PDT (Figure 1).

The foveal retinal thickness and choroidal thickness measurements at one, three, six, and 12 months after treatment are shown in Figure 2.

The central retinal thickness and choroidal thickness were significantly decreased at each measurement as compared with those before treatment. The rate of dry macula at one month after treatment was 89%, and that at 12 months was also 89%. None of the eyes showed macular atrophy at any of the post-treatment measurements.

Three patients required additional treatment; two were treated one month after 2/3PDT and the other eight months later. One of these three patients required the subsequent addition of aflibercept, administered five times, while the other two received aflibercept only once each (Table 2).

Univariate analyses were performed to identify factors such as age, sex, reflective error, steroid use, smoking, disease duration, presence of CVH and drusen, but none of the examined factors showed significant associations with retreatment (Table 3).

None of the 27 eyes developed subretinal hemorrhage after 2/3PDT.

A representative case is shown in Figure 3.

## 4. Discussion

Smretschnig et al. reported 17 eyes developing CNV after chronic CSC, i.e., PNV, treated with VEGF followed by half-dose PDT. During the year of follow-up, 47% of cases required additional VEGF therapy [9]. Hata et al. reported that VEGF levels in the aqueous humor of eyes with PNV were lower than those of eyes with tAMD [12]. This suggests anti-VEGF treatment is possibly not effective for PNV. Matsumoto et al. also analyzed the treatment outcomes for PNV; they divided patients into two groups, with polyps (PCV) and without polyps, and treated them with anti-VEGF agents. The PCV group required fewer intravitreal injections than the PCV group without polyps, which we defined as PNV. These results suggest that PNV (excluding PCV) may be less responsive to anti-VEGF injection than PCV. In addition, Matsumoto et al. reported the one-year results of half-dose PDT and VEGF combination therapy for PNV without polyps [7]. Seventeen of 21 eyes (81%) showed no recurrence, but of the four patients with recurrence, three required a single additional administration of combination therapy and the other required three additional concomitant therapies.

Choroidal vascular hyperpermeability is known to be a cause of CSC [14], and reduced-dose PDT is reportedly effective for this form of CSC [15,16]. Since we excluded eyes with PCV from this study and there would presumably be many cases in which CNV arose from CSC with a thickened choroid [17,18], we anticipated that 2/3PDT would be effective. On the other hand, Lee et al. examined the one-year results of full-dose PDT for thickened choroid Type 1 CNV, and found that 40% of eyes showed relapse at one year [13]. We also performed reduced-dose PDT, but the recurrence rate at one year was 11%. This might reflect differences in the origin of PNV between our patient population and that of the study by Lee et al. The average age of our patients was 58 years, while that in the study by Lee and colleagues was 66 years on average, and their CVH rate was 46% compared to 100% in our patients. These results indicate that PDT is more effective for PNV arising from CSC with marked CVH than for other forms of PNV.

Recently, the one-year results of combined treatment with ranibizumab and full-dose PDT for PNV without PCV was reported [8]. During a year of follow-up, as many as 45% of patients were administered additional injections, requiring anti-VEGF therapy 3.9 times on average. There was a major difference in the number of additional injections given during the one-year period after initial treatment as compared with our study, with 11% of patients being administered additional injections, requiring anti-VEGF therapy 0.25 times on average during the one-year follow-up period. Reasons for this may include the use of ranibizumab and the exclusion of cases with a CSC history. Ranibizumab use might account for this difference, based on reports of lower dry macula rates as compared with aflibercept [10]. Furthermore, since only cases without a medical history of CSC were enrolled in their study, the probability of PDT being effective for CVH was quite low and the lesions treated may have been mature Type 1 CNV. Yanagi et al. reported that treating CVH achieved better results when combinations of anti-VEGF agents and PDT were administered for PCV [19].

Macular atrophy reportedly occurs when PDT is performed at conventional doses. Recently, Miyata et al. reported the five-year results of therapy combining anti-VEGF drugs with PDT for PCV. The combination therapy with PDT was more likely to result in macular atrophy than that with anti-VEGF alone [20]. Son et al. compared half-dose versus full-dose PDT for CSC and found no difference in macular atrophy at three years between the two groups [21]. We previously reported that PDT monotherapy was administered for PCV with a three-year follow-up. Eight of the 43 eyes developed macular atrophy, which was associated with decreased visual acuity [22]. On the basis of these observations, we devised a regimen with an intermediate dose, 2/3PDT, which while being less effective than full-dose PDT is also less likely to cause macular atrophy. The 12-month rate of dry macula was good at 89%, and visual acuity was also significantly increased. Furthermore, none of the treated eyes developed macular atrophy. Five additional treatments with an anti-VEGF injection were required in one patient, while two others showed a dry macula after only one additional injection. Although there were no significant differences in the characteristics of the three patients who relapsed, they were more elderly than the entire group, whose mean age was 58.9 years, and also two of three patients had drusen. Patients 1 and 3 had rather thin choroids, 196 and 201 μm, respectively, as compared to the mean choroidal thickness of 307 μm of the entire patient group. The univariate analysis for factors of additional treatment (Table 3) showed that the sex might be a confounding factor. The study is not powered to assess this given the greater number of men than women included. The cause is unknown and needs to be studied further.

These results suggest 2/3PDT to be effective for CVH in PNV without polyps.

This study has several limitations. The number of treated patients was too small and the follow-up period was too short to clarify macular atrophy. This study was retrospective in nature and we did not compare patients with and without anti-VEGF treatment. Further study, with a larger patient population, is needed to clarify the impacts of these factors.

Herein, we have shown 2/3PDT to be effective for treating PNV without polyps.

## Figures and Tables

**Figure 1 jcm-10-02168-f001:**
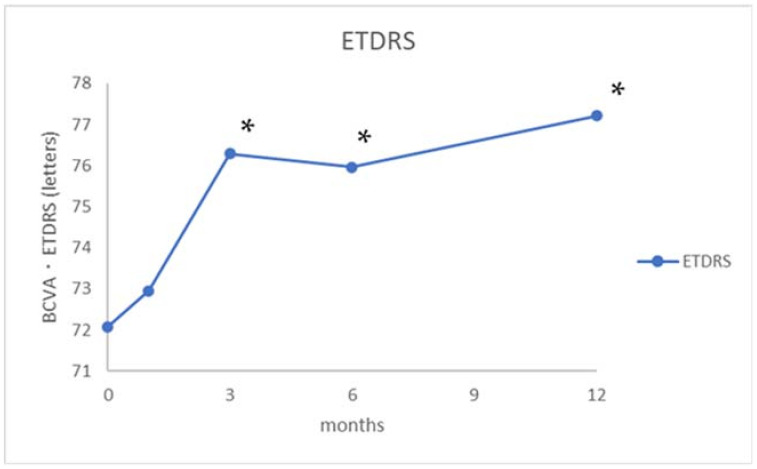
Changes in best corrected visual acuity (BCVA) (ETDRS: Early Treatment Diabetic Retinopathy Study) during the 12-month study period. * *p* < 0.05.

**Figure 2 jcm-10-02168-f002:**
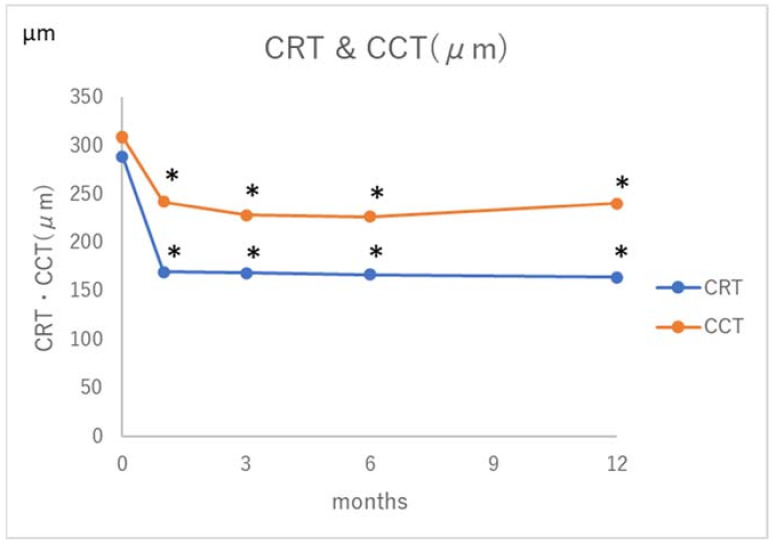
Central macular retinal thickness (CRT) & central macular choroidal thickness (CCT) in eyes with pachychoroid neovasculopathy treated with 2/3PDT. * *p* < 0.05.

**Figure 3 jcm-10-02168-f003:**
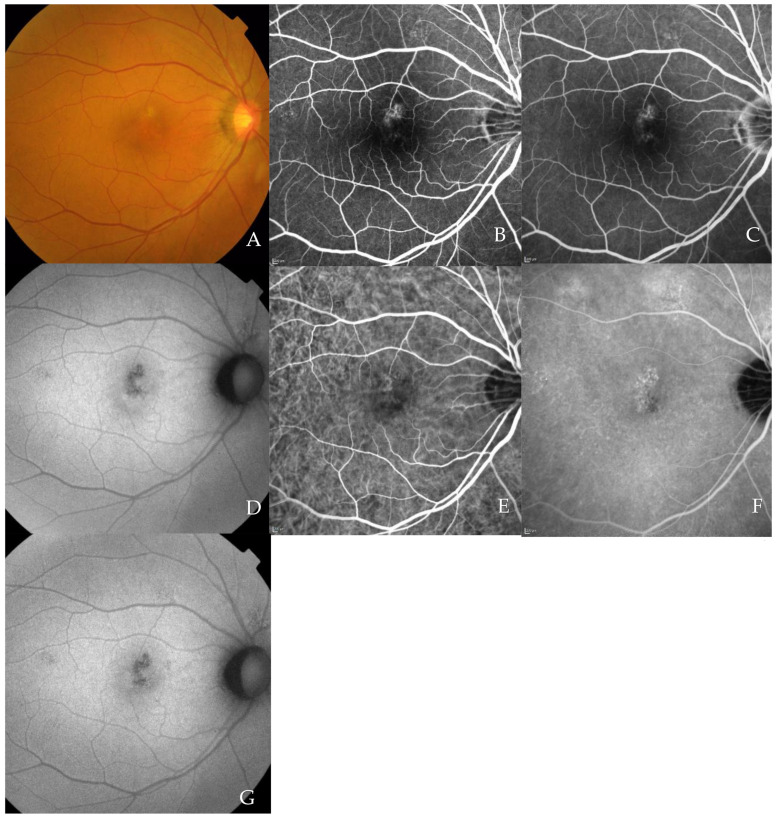
Multimodel imaging of 63-year-old PNV patient. Color fundus graph at baseline (**A**). Fluorescein angiography in the early phase showed a hyperfluorescent area corresponding to PNV (**B**). Fluorescein angiography in the late phase showed the same hyperfluorescent area as in B (**C**). Fundus autofluorescence showed a hypofluorescent lesion in the macula at baseline (**D**). Indocyanine green angiography in the early phase showed a hyperfluorescent area corresponding to PNV (**E**). Indocyanine green angiography in the late phase showed a hyperfluorescent area corresponding to PNV, and choroidal hyperpermeability was seen in the arcade; arrows (**F**). Fundus autofluorescence at 12 months after PDT (**G**). Optical coherence tomography (OCT) at baseline. Serous retinal detachment, irregular retinal pigment epithelium; dotted arrow and pachyvessels were seen; arrow (**H**). OCT one month after PDT. Dry macula (**I**). OCT 12 months after PDT. Dry macula with no additional injection (**J**). 3 mm × 3 mm OCT angiography (OCTA) with B-scan at baseline (**K**). OCTA 1 month after PDT (**L**). OCTA 12 months after PDT. Network vessel were getting large (**M**).

**Table 1 jcm-10-02168-t001:** Baseline characteristics of subjects (means ± SD).

*n*	27
Sex (male/female)	20:7
Age (y)	58.9 ± 9.3
Refractive error (D)	−1.39 ± 2.9
Steroid use	11%
Smoking	15%
Disease duration (months)	42.8 ± 39.9
Drusen	22%
CVH	100%
CRT (μm)	294 ± 74.8
CCT (μm)	307 ± 109.7
BCVA (ETDRS; letters)	72.1 ± 12.6

*n*: number, CVH: choroidal vascular hyperpermeability, CRT: central macular retinal thickness, CCT: central macular choroidal thickness, BCVA: Best corrected visual acuity, ETDRS: Early Treatment Diabetic Retinopathy Study.

**Table 2 jcm-10-02168-t002:** Patients who needed additional treatment at baseline.

	Patient 1	Patient 2	Patient 3
Age (years)	74	71	60
Sex	male	female	female
Refractive error (D)	0.5	2.25	−5
Steroid use	−	−	−
Smoking	−	−	−
Disease duration (months)	3	4	24
Drusen	+	+	−
Baseline CRT(μm)	256	313	439
Baseline CCT(μm)	196	477	201
Baseline visual acuity ETDRS (letters)	67	77	75
ETDRS (letters) at 12 months	79	82	80
Additional injection (times)	5	1	1
Recurrence (months after PDT)	1	8	1

CRT: central retinal thickness, CCT: central choroidal thickness; ETDRS: Early Treatment Diabetic Retinopathy Study, PDT: photodynamic therapy.

**Table 3 jcm-10-02168-t003:** Univariate analysis for factors of additional treatment.

	*p*-Value
Age	0.065
Sex	0.088
Refractive error	0.680
Steroid use	0.516
Smoking	0.444
Disease duration	0.135
Drusen	0.050
Baseline CRT	0.308
Baseline CCT	0.787
Baseline visual acuity	0.905

## Data Availability

The datasets of this study are available from the corresponding author upon reasonable request.

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
