# Peer review of "Two-Thirds Dose Photodynamic Therapy for Pachychoroid Neovasculopathy"

_jcm, 2021, doi:10.3390/jcm10102168_

Round 1
Reviewer 1 Report
The Authors have written an interesting article on an important topic because treatment with intravitreal injection only does not seem to be sufficently effective in this pathology. I would suggest to highlight the importance of two-thirds photodynamic therapy in order to reduce RPE impairment and damage in these patients, adding also autofluorescence images before and after treatment.
Author Response
1) The Authors have written an interesting article on an important topic because treatment with intravitreal injection only does not seem to be sufficiently effective in this pathology. I would suggest to highlight the importance of two-thirds photodynamic therapy in order to reduce RPE impairment and damage in these patients, adding also autofluorescence images before and after treatment.
⇒Thank you for your helpful comment. We added the FAF images as “G “of 12 months after PDT in Figure 3.
Reviewer 2 Report
Ms. Ref. No.: jcm-1144886
Title: Two-thirds Photodynamic Therapy for Pachychoroid Neovas-2 culopathy
Koji Tanaka, Ryusaburo Mori, Yu Wakatsuki, Hajime Onoe, Akiyuki Kawamura and Hiroyuki Nakashizuka
Overview and general recommendation:
In this manuscript, the authors present the results obtained applying a reduced PDT protocol to treat Pachychoroid neovasculopathy (PNV). In particular, they suggest the 2/3 PDT treatment that consist in administrating a verteporfin injection at 4 mg/m2 of body surface area (2/3 of the usual dose) and then underwent PDT with a light fluence of 50 J/cm2 using a Visulas PDT system 690S (Carl Zeiss Japan) for 83 seconds. The authors report the treatment outcomes of 27 eyes of 27 patients who underwent this novel schema of PDT and were followed up for 1 year.
Nowadays, various studies were conducted, but no optimally PDT effective treatments for PNV have yet been established. Although the standard protocol provides for an injection of verteporfin at 6 mg / m2 of body surface area and then subjected to PDT with a slight fluence of 50 J / cm2, many papers report the effectiveness of modified protocols. Smretschnig et al in 2016 presented the results obtained with a PDT protocol in which only half of the fluence was used.
I find the work well written, the research well designed and the results are promising although further studies are needed to understand the applicability of this novel protocol in clinical practice.
Major concerns:
In the title and in the body of the text the authors speak of 2/3 PDT protocol without ever underlining that 2/3 comes from a reduction in the dose of the drug and not from a decrease in the energy administered. In my opinion, this concept should also be clearly stated in the title and abstract and not only in the materials and methods section.
Minor concerns:
A parenthesis is missing in the abstract on line 15.
In table 1 it should be specified that the data are reported as mean and SD
Although not significant, the results of the logistic regression should be reported. The lack of significance could be due to the low number of the sample examined, and to the too many variables considered. It would have been better to first perform a univariate analysis to highlight any significant variables and then proceed with the logistic regression.
Author Response
Major concerns: In the title and in the body of the text the authors speak of 2/3 PDT protocol without ever underlining that 2/3 comes from a reduction in the dose of the drug and not from a decrease in the energy administered. In my opinion, this concept should also be clearly stated in the title and abstract and not only in the materials and methods section.
⇒Thank you for your helpful comment. We added the “dose” in the title, so that everyone can understand the protocol of PDT.
Minor concerns:
A parenthesis is missing in the abstract on line 15.
⇒Thank you for your suggestion. We added the parenthesis on line 15.
In table 1 it should be specified that the data are reported as mean and SD.
⇒We added the “ mean and SD ” in Table 1.
Although not significant, the results of the logistic regression should be reported. The lack of significance could be due to the low number of the sample examined, and to the too many variables considered. I would have been better to first perform a univariate analysis to highlight any significant variables and then procced with the logistic regression.
⇒We added the univariate analysis in Table 4. However, there was no significant difference. We also added the sentence in Discussion part line 225. “and also two of three patients had drusen.”